# The Induction Mechanism of Ferroptosis, Necroptosis, and Pyroptosis in Inflammatory Bowel Disease, Colorectal Cancer, and Intestinal Injury

**DOI:** 10.3390/biom13050820

**Published:** 2023-05-11

**Authors:** Ping Zhou, Shun Zhang, Maohua Wang, Jun Zhou

**Affiliations:** 1Department of Anesthesiology, The Affiliated Hospital of Southwest Medical University, Luzhou 646000, China; 20220299120536@stu.swmu.edu.cn (P.Z.); 20200299120542@stu.swmu.edu.cn (S.Z.); wangmaohua@swmu.edu.cn (M.W.); 2Anesthesiology and Critical Care Medicine Key Laboratory of Luzhou, Luzhou 646000, China

**Keywords:** ferroptosis, necroptosis, pyroptosis, inflammatory bowel disease, colorectal cancer, intestinal injury

## Abstract

Cell death includes programmed and nonprogrammed cell death. The former mainly includes ferroptosis, necroptosis, pyroptosis, autophagy, and apoptosis, while the latter refers to necrosis. Accumulating evidence shows that ferroptosis, necroptosis, and pyroptosis play essential regulatory roles in the development of intestinal diseases. In recent years, the incidence of inflammatory bowel disease (IBD), colorectal cancer (CRC), and intestinal injury induced by intestinal ischemia–reperfusion (I/R), sepsis, and radiation have gradually increased, posing a significant threat to human health. The advancement in targeted therapies for intestinal diseases based on ferroptosis, necroptosis, and pyroptosis provides new strategies for treating intestinal diseases. Herein, we review ferroptosis, necroptosis, and pyroptosis with respect to intestinal disease regulation and highlight the underlying molecular mechanisms for potential therapeutic applications.

## 1. Introduction

Cell death is the termination of cell life phenomena that often occur in normal tissues and is a necessary life process to maintain tissue function and morphology. Previous studies have indicated that cell death includes two pathways: apoptosis and necrosis. In recent years, many studies have shown that cell death includes other types, such as ferroptosis, necroptosis, and pyroptosis [1].

Ferroptosis is a unique form of iron-dependent nonapoptotic regulated cell death, proposed in 2012. It is pathologically characterized by the accumulation of intracellular iron-dependent lipid peroxidation [2]. Excess iron generates a large number of reactive oxygen species (ROS) through the Fenton reaction, while the depletion of reduced glutathione (GSH) and/or the inhibition of glutathione peroxidase 4 (Gpx4) increases the accumulation of intracellular ROS [2,3]. When the level of ROS exceeds the scavenging level of the body’s antioxidant system, it oxidizes the unsaturated fatty acids on the cell membranes and organelle membranes to form lipid peroxides, directly or indirectly damaging the cell structure and function, resulting in cell damage or death [4]. Small molecular substances, such as erastin, Ras-selective lethal 3 (RSL3), salazosulfapyridine, and butylthiocyanine sulfoxide imine, can induce ferroptosis [5,6,7]. Additionally, some drugs, such as sorafenib, as well as artemisinin and its derivatives, can induce ferroptosis [7,8]. Conversely, ferrostatin-1 (Fer-1), liproxstatin-1 (Lip-1), deferoxamine (DFO), and vitamin E inhibit ferroptosis [9]. To date, several molecules, including GPX4, p53, solute carrier family 7 member 11 (SLC7A11), acyl-CoA synthase long-chain family member 4 (ACSL4), NADPH oxidase (NOX), and nuclear factor E2-related factor 2 (NRF2), have been identified to positively or negatively regulate ferroptosis [10,11,12,13,14].

Necroptosis was first proposed in 2005 as a new mode of programmed cell death [15]. It is characterized by a serine/threonine protein kinase 1/3 (RIPK1/RIPK3)-mediated phosphorylation signaling pathway activating mixed lineage kinase domain-like protein (MLKL/pMLKL). Death receptors activate RIPK1 and RIPK3, and RIPK3 promotes the phosphorylation of MLKL, leading to plasma membrane destruction and the release of damage-associated molecular patterns (DAMPs) and cellular contents to mediate a series of immune and inflammatory responses [16,17,18,19].

The term pyroptosis, first coined in 2001, is a form of programmed cell death that accompanies inflammation [20] and comprises the gasdermin family proteins (GSDMs). This family mediates cellular pore formation, causing cell swelling and lysis and releasing cellular contents that activate a robust inflammatory response [21]. The mechanisms of pyroptosis mainly include inflammasome-activated pyroptosis and noninflammasome-activated pyroptosis.

Intestinal epithelial cells are an essential functional barrier of the intestine. Under the invasion of pathogens, intestinal epithelial cells die to maintain intestinal epithelial function, continuous renewal, and tissue homeostasis. However, intestinal epithelial cell death imbalance elevates intestinal permeability and barrier dysfunction, leading to various acute and chronic intestinal diseases, such as inflammatory bowel disease, colorectal cancer, and intestinal injury caused by intestinal ischemia—reperfusion [22,23]. The physiological and pathological effects of cell death in various diseases have been studied extensively [24,25,26]. However, there are still gaps in the basic knowledge of the mechanism of intestinal epithelial cell death in intestinal diseases. Notably, ferroptosis, necroptosis, and pyroptosis are indispensable during intestinal development and homeostasis and occur throughout life. In conclusion, there is an urgent need to understand cell death mechanisms in intestinal diseases to develop promising therapeutic strategies.

Taken together, ferroptosis, necroptosis, and pyroptosis play vital roles in the pathophysiological process, providing novel approaches for treating intestinal diseases. Therefore, this review systematically overviews the mechanisms of ferroptosis, necroptosis, and pyroptosis. We also investigated their roles in several intestinal disorders, including IBD, CRC, intestinal injury induced by intestinal I/R, sepsis, and radiation therapy.

## 2. Mechanisms of Ferroptosis, Necroptosis, and Pyroptosis

### 2.1. Mechanism of Ferroptosis

#### 2.1.1. GSH/Gpx4–Lipid Peroxidation Pathway

Glutathione is a tripeptide compound composed of glutamate, cysteine, and glycine and is an essential component of the antioxidant system in the body. Depleting GSH leads to redox imbalance in the body and the accumulation of considerable free oxygen radicals, which in turn causes lipid peroxidation and cell death [27] (Figure 1). The synthesis of GSH requires glutathione synthetase (GSS) and glutamate–cysteine ligase (GCL), and GPX4 is an essential regulator of ferroptosis [28]. The ferroptosis-inducer RSL3 induces ferroptosis by directly inhibiting GPX4 activity through covalent binding to selenocysteine, the active site of GPX4, which is self-inactivated under GSH depletion [3,4]. FIN56, a specific inducer of ferroptosis, also causes GPX4 degradation and initiates iron death [29]. Cysteine is the rate-limiting substrate for GSH biosynthesis; it is produced from the dipeptide cysteine imported by the cell surface cysteine/glutamate reverse transport system Xc^–^ or from methionine via the trans-sulfuration pathway. Cysteine deficiency or Xc^–^ inhibition decreases GSH levels and GPX4 activity and increases ROS levels, thereby promoting ferroptosis [30,31]. System Xc^–^ is an essential target in the regulation of ferroptosis. Studies have shown that erastin, sulfasalazine (SAS), and sorafenib can bind to related transport proteins, block the transport function of system Xc^–^, and induce cellular ferroptosis [2,32,33]. Reportedly, p53 affects the activity and ferroptosis sensitivity of the Xc^–^ system in cancer cells by transcriptionally downregulating the expression of the functional subunit SLC7A11 in the Xc^–^ system [11]. Furthermore, glutaminase, a transcriptional target of the tumor suppressor p53, inhibits ferroptosis by limiting dipeptidyl-peptidase-4-mediated lipid peroxidation [34]. CD8+ T cells promote ferroptosis in tumor cells by releasing interferon-gamma (IFN-γ) and suppressing the expression of SLC3A2 and SLC7A11 in tumor cells [35]. The core of ferroptosis is the reaction of ROS with the polyunsaturated fatty acids (PUFAs) of lipid membranes, causing lipid peroxidation. Among the different PUFAs, arachidonic acid and adrenergic acid are critical components inducing ferroptosis [10]. Free PUFAs need to be esterified and incorporated into membrane phospholipids (PLs) via ACSL4 and lysophosphatidylcholine acyltransferase 3 (LPCAT3) to effectuate lipid peroxidation. In addition, thiazolidinediones attenuate ferroptosis by inhibiting the activity of ACSL4 and PUFA peroxidation [10]. This phenomenon suggests that ACSL4 and LPCAT3 play an essential role in ferroptosis. Lipoxygenase also plays a critical role in developing ferroptosis, and lipoxygenase (LOX) inhibitors, such as zileuton and baicalein, inhibit LOX-catalyzed lipid peroxidation [36].

#### 2.1.2. Iron Metabolism Imbalance Pathway

Iron is an indispensable trace element in the growth and development of humans. The primary mechanism for the biotoxicity of iron ions in the classical Fenton reaction between Fe^3+^ and Fe^2+^ generates ROS that can damage cellular proteins, membrane lipids, and DNA [2] (Figure 1). In addition, iron participates in enzymatic lipid reactions as a cofactor of LOX [37]. Therefore, regulating iron metabolic processes may be a new target and direction for regulating ferroptosis. DFO and deferiprone chelate intracellular iron to inhibit erastin-induced ferroptosis. Extracellular iron forms a complex with circulating transferrin (TF) that binds to the TF receptor (TFR) on the cell membrane and is transported into the cell, wherein excess iron is stored in ferritin or transported into the circulation via the iron pump SLC11A2/divalent metal transporter 1 (DMT1) [38]. Ferroportin (FPN) mediates the cellular export of iron and is located on the basolateral membrane of intestinal epithelial cells. FPN-mediated iron transport is tightly regulated by hepcidin encoded by the *HAMP* gene in hepatocytes, which binds to FPN to inhibit the cellular export of iron [39]. Decreased iron storage or increased iron uptake causes iron overload and triggers ferroptosis, while ferritin-selective autophagy increases susceptibility to ferroptosis [40]. The frataxin protein is localized to the mitochondria and involved in the biosynthesis of iron–sulfur clusters [41]. Decreased frataxin expression can lead to iron accumulation at the mitochondrial level. On the other hand, the inhibition of frataxin expression accelerates free iron accumulation, promotes lipid peroxidation, and leads to ferroptosis [42], while overexpression blocks erastin-induced ferroptosis. Therefore, the frataxin protein is considered a key regulator of ferroptosis. Other proteins, such as heat shock protein beta-1 (HSPB1) and CDGSH iron–sulfur structural domain 1 (CISD1), affect iron metabolism and susceptibility to ferroptosis [43,44]. Therefore, maintaining iron metabolism homeostasis is critical for ferroptosis homeostasis.

#### 2.1.3. Other Mechanisms of Ferroptosis

In addition to the two significant mechanisms described above, other mechanisms may also be closely related to the occurrence of ferroptosis (Figure 1). Several studies have shown that the p53–SLC7A11P53 axis and the production of coenzyme ubiquinone via the mevalonate pathway and the GCH1–BH4 axis may be linked to the ferroptosis mechanism [29,45]. FIN56 decreases the coenzyme Q10 (CoQ10) level in the mevalonate pathway by activating squalene synthase, thereby reducing the antioxidant capacity of cells [29]. In addition, many studies in recent years have shown that the mitochondria play a role in promoting and inhibiting ferroptosis. The promoting role is related to voltage-dependent anion channels and cardiolipin, and the inhibitory role is associated with CDGSH iron–sulfur domain 1 and mitochondrial ferritin [44,46,47,48]. According to reports, iron ions and metabolites in eukaryotic cells can be transported through membrane pore proteins on the outer mitochondrial membrane. Ferroptosis inducers can bind to membrane pore protein 2/3, thus altering mitochondrial membrane permeability and reducing the sensitivity of channels to iron ions; this process causes mitochondrial dysfunction and the release of many oxidative substances, ultimately leading to the onset of ferroptosis [49]. However, some researchers still believe that the role of mitochondria in ferroptosis is limited [2,50]. Therefore, exploring the role of mitochondria in ferroptosis will help to understand ferroptosis-related diseases and find new therapeutic targets.

### 2.2. Mechanism of Necroptosis

Necroptosis is closely associated with death receptors (TNFR1, CD95, and TRAIL-R), Toll-like receptors (TLR3 or TLR4), or cytoplasmic Z-DNA/Z-RNA sensing receptor Z-DNA binding protein 1 (ZBP1/DAI/DLM-1) [51]. After the interaction of tumor necrosis factor-alpha (TNF-α) with TNF receptor 1 (TNFR1), TNFR1 starts to recruit downstream protein molecules, such as TNFR1-associated death domain protein (TRADD), cellular inhibitors of apoptosis proteins (cIAPs), TNFR-associated factor 2/5, and linear ubiquitin chain assembly complex proteins, to form complex I. In this complex, RIPK1 is polyubiquitinated and activates the nuclear factor-kappa B (NF-κB) and mitogen-activated protein kinase (MAPK) signaling pathways, inhibiting the activation of caspase-8 and promoting cell survival [52]. When caspase-8 and/or the apoptotic protein cIAP are removed, these receptors activate RIPK1/3 to induce necroptosis directly [53]. Moreover, when caspase-8 is inhibited, or its activity level is low, RIPK1 recruits RIPK3 and interacts via the heteroamyloid RIPK1/3 RHIM–RHIM structural domain to trigger the formation of the RIPK1–RIPK3 cell death platform, also known as the necrosome. The subsequent RIPK3-mediated phosphorylation of the necroptotic executor MLKL triggers its oligomerization and membrane binding, leading to plasma membrane damage and potential DAMP release [54] (Figure 2). Due to membrane damage, potassium efflux further activates the NLRP3 inflammasome through NIMA-related kinase 7 (NEK7), thereby increasing the release of inflammatory mediators and cellular content [55,56]. Furthermore, the inhibition of RIPK1 by the necroptosis inhibitor necrostatin-1 (Nec-1) inhibits necroptosis by preventing the formation of complex IIb, suggesting an essential role of RIPK1 in necroptosis [15].

### 2.3. Mechanisms of Pyroptosis

#### 2.3.1. Inflammasome-Activated Pyroptosis Pathway

The inflammasome is a multimolecular complex activated in response to host infection by pathogens or internal damage factors, initiating adaptive immune and inflammatory responses. Inflammasomes include classical and nonclassical inflammasomes. The former mainly includes NLRP3, NLRC4, melanoma 2 (AIM2), NLRP1, and pyrin inflammasomes [57], while the latter mainly refers to the lipopolysaccharide (LPS) of Gram-negative bacteria [58]. Most inflammasomes are composed of three components: leucine-rich repeat proteins (NOD-like receptors (NLRs)), adapter apoptosis-associated spot-like proteins containing (ASC) the cysteine aspartase recruitment domain (CARD), and procaspase-1. Pathogen-associated molecular patterns (PAMPs) and DAMPs are recognized by the pattern recognition receptor (PRR, also known as the inflammasome sensor) [59,60]. After cell stimulation by signaling molecules from pathogenic microorganisms (such as bacteria, fungi, and viruses), the PRR assembles with procaspase-1 and ASC to form inflammasomes [61,62]. Subsequently, caspase-1 is activated and forms a dimer, which becomes a mature cleaved caspase-1 [63]. Interestingly, activated caspase-1 cleaves GSDMD (a member of the gasdermin protein family consisting of more than 500 amino acids) to form a 22 kDa C-terminus (GSDMD-C) and a 31 kDa N-terminus (GSDMD-N). GSDMD-N penetrates and forms a hole in the cell membrane, causing cell swelling and pyroptosis. On the other hand, the activated caspase-1 recognizes the inactive IL-β and IL-18 precursors and cleaves them into mature IL-1β and IL-18, which are released through the pore formed by GSDMD, leading to pyroptosis [64,65] (Figure 3). Interestingly, the Gram-negative bacterial cell wall component LPS is recognized by caspase-4 and caspase-5 in human cells and caspase-11 in mouse cells [58]. Then, caspase-4/-5/-11 directly cleaves GSDMD and triggers pyroptosis [66], while amino-terminal GSDMD-N activates the NLRP3 inflammasome and mediates IL-1β/IL-18 maturation and secretion via the NLRP3–caspase-1 pathway [67].

#### 2.3.2. Noninflammasome-Activated Pyroptosis Pathway

Pyroptosis can occur even in the absence of inflammasomes (Figure 3). Additionally, chemotherapeutic drugs activate caspase-3, which shears GSDME at the Asp270 site, forming the GSDME-N-terminus and ultimately leading to tumor pyroptosis [68]. In addition, in mouse macrophages, *Yersinia pestis* infection inhibited the transforming growth factor β (TGF-β)-activated kinase 1 (TAK1) activity, which activated the cleavage of GSDMD by caspase-8 [69,70]. Another study found that antibiotic chemotherapeutic drugs induce caspase-8 activation, which then induces tumor cell pyroptosis by cleaving GSDMC [71]. In addition to apoptotic caspase-mediated pyroptosis, neutrophil elastase (ELANE) in neutrophils activates GSDMD by the cleavage at cysteine 268 in a caspase-1/11-independent manner, triggering neutrophil pyroptosis [72]. Zhou et al. [73] first demonstrated that granzyme A in cytotoxic lymphocytes hydrolyzes GSDMB in tumor cells at nonaspartate sites to induce pyroptosis. Moreover, Zhang et al. [74] demonstrated that granzyme B, a granzyme in natural killer (NK) cells and cytotoxic T lymphocytes (CTL), directly cleaves GSDME and induces pyroptosis. These studies suggest that the role of noninflammasome activation mechanisms in pyroptosis is gaining increasing attention.

## 3. Role of Ferroptosis, Necroptosis, and Pyroptosis in Intestinal Diseases

### 3.1. Role of Ferroptosis in Intestinal Diseases

#### 3.1.1. Ferroptosis and IBD

Accumulating evidence suggests that ferroptosis is associated with the pathogenesis of IBD (Table 1). Blocking the ferroptosis process alleviates dextran sodium sulfate (DSS)-induced colitis [75]. Xu et al. [76] showed that ferroptosis is associated with CD, and the ferroptosis inhibitor ferrostatin-1 attenuates the pathological phenotype of trinitrobenzenesulfonic acid (TNBS)-induced CD-like colitis in mice. Heme oxygenase 1 (HO-1) is significantly upregulated in a DSS-induced model of experimental colitis, exerting anti-inflammatory and antioxidant properties [77]. Triantafillidis et al. [78] found that the overactivation of Nrf2–HO-1 induced ferroptosis by disturbing the balance of iron ion metabolism to participate in DSS-induced UC, while ferrotitain-1 administration ameliorated colitis through the Nrf2–HO-1 signaling pathway [79]. In addition, furin protected epithelial cells from DSS-induced ferroptosis-like cell damage and alleviated experimental colitis by activating the Nrf2–Gpx4 signaling pathway [80]. PUFAs are critical factors in the ferroptosis process and can trigger GPX4-restricted mucosal inflammation, similar to some aspects of human CD [81]. Reportedly, CD44 and MUC1 may be ferroptosis-related markers in UC [82]. In addition, the treatment of OTSSP167 (a selective inhibitor of MELK) attenuates intestinal tissue damage and inflammation by modulating gut microbial composition and inhibiting ferroptosis [83]. Interestingly, research on phytobioactive agents in herbal medicine has also made progress with respect to IBD treatment [84,85]. These findings may provide new perspectives for developing therapeutic strategies for IBD.

#### 3.1.2. Ferroptosis and CRC

Several studies have demonstrated that ferroptosis can limit the migration, invasion, and proliferation of CRC (Table 1). Resibufogenin inhibits the growth of CRC cells by inducing ferroptosis, suggesting that the inhibition of ferroptosis is closely associated with the proliferation of CRC cells [86]. RSL3-induced ferroptosis in CRC cells is associated with cancer progression [6]. The combination of ferroptosis inhibitors with other agents or drugs can improve the safety of CRC treatment. Lee et al. [87] showed that the combination of ferroptosis drugs and TNFR apoptosis-inducing ligands inhibits the progression of CRC. Many microRNAs (miRNAs) have been directly or indirectly associated with oncogenes, especially in CRC. miR-19a, one of the most critical miRNAs, inhibits ferroptosis in CRC by suppressing the expression of iron response element-binding protein 2 (IREB2) [88]. miR-15a-3p overexpression inhibited GPX4 by binding to the 3′-untranslated region of GPX4, resulting in increased ROS levels, intracellular Fe^2+^ levels, and malondialdehyde accumulation in vitro and in vivo; this phenomenon suggests that miR-15a-3p inhibits the expression of GPX4 in CRC cells to positively regulate ferroptosis [89]. In addition, ferroptosis affects the resistance and chemosensitivity of CRC to chemotherapeutic agents. Andrographis activates ferroptosis and inhibits β-catenin/Wnt signaling pathway-mediated chemosensitization in CRC [90], providing a novel target to overcome resistance to ferroptosis activators. Conversely, FAM98A promotes resistance to 5-fluorouracil (5-Fu) in CRC by inhibiting ferroptosis, suggesting that ferroptosis can positively and negatively regulate resistance to CRC chemotherapeutic agents [91]. Recently, a study showed significantly increased TP53-induced glycolysis and apoptosis regulator (TIGAR) expression in CRC tissues. At the same time, its depletion promoted lipid peroxidation via increased ROS production and the ROS-/AMPK-mediated downregulation of SCD1 expression [92]. The study also determined that blocking this signaling pathway may promote ferroptosis in human CRC. Ferroptosis-related genes (FRGs) have been investigated, and a prognostic correlation model for CRC related to ferroptosis has been developed [93]. Therefore, identifying ferroptosis-related genes may provide a significant prognostic and therapeutic basis for diagnosing and treating CRC patients. Some studies have shown that plant compounds induce ferroptosis in CRC cells and inhibit tumor growth [94,95]. Recently, auriculasin was also found to promote CRC cell death via ROS production [96]. Together, these studies suggest that ferroptosis is critical for regulating the growth and proliferation of CRC cells. The pharmacological activation of ferroptosis-related signaling pathways is a promising strategy for CRC treatment.

#### 3.1.3. Ferroptosis and Intestinal Injury

ROS production and lipid peroxidation are the main steps of ferroptosis and major contributors to intestinal I/R injury [97]. Accumulating evidence suggests that ferroptosis is associated with the pathophysiological process of intestinal I/R injury and is a potential target for its treatment (Table 1). Iron chelators attenuate I/R injury in a rat model, suggesting that ferroptosis is closely related to intestinal I/R injury [98]. Nrf2 is a crucial regulator of intracellular oxidative homeostasis and acts as an antioxidant. Dong et al. [80] demonstrated that furin inhibits intestinal epithelial cell damage and alleviates experimental colitis by activating the Nrf2–Gpx4 signaling pathway. Tert-butylhydroquinone (TBHQ) inhibits ferroptosis and attenuates 5-Fu-induced intestinal epithelial cell injury by activating Nrf2 [99]. Radiation disrupts the intestinal mucosal barrier by promoting gut microbial dysbiosis, and ferroptosis plays a significant role in promoting ionizing radiation-induced intestinal damage [100]. Dar et al. [101] found enhanced gut colonization with *Pseudomonas aeruginosa* (PAO1)-induced ferroptosis in host cells and significantly reduced survival in irradiated mice. In contrast, baicalein significantly reduced PAO1 colonization and inhibited the occurrence of ferroptosis in intestinal epithelial cells. In addition, recent studies have shown that the green tea derivative (-)-epigallocatechin-3-gallate (EGCG) prevents radiation-induced intestinal injury by scavenging ROS and inhibiting apoptosis and ferroptosis through the Nrf2 signaling pathway, which might be a promising therapeutic strategy for alleviating radiation-induced intestinal injury [102]. D-prostaglandin E2 accelerates the recovery of chemotherapy-induced intestinal injury by upregulating the expression of cyclin D and is an effective protective agent against intestinal injury [103]. Previous studies have shown that some compounds and agonists can inhibit ferroptosis and alleviate intestinal damage, for example, polyphenols sourced from *Ilex latifolia thunb* (PIT) [104] and an aldehyde dehydrogenase 2 (ALDH2) agonist (Alda-1) [105]. These findings suggest that ferroptosis inhibition could be a new method to treat intestinal damage.

**Table 1 biomolecules-13-00820-t001:** Mechanisms of ferroptosis in inflammatory bowel diseases, colorectal cancer, and intestinal injury.

Compound/Target	Model	Effect	Mechanism	Ref.
Fer-1	TNBS-induced colitis mice	Inhibition	Ferrostatin-1 alleviates TNBS-induced colitis via the inhibition of ferroptosis.	[76]
Fer-1/Lip-1/DFP	DSS-induced colitis mice	Inhibition	Ferroptosis-mediated DSS-induced ulcerative colitis is associated with the Nrf2–HO-1 signaling pathway.	[79]
Furin	DSS-induced colitis mice; NCM460 cells	Inhibition	Furin inhibits epithelial cell injury and alleviates experimental colitis by activating the Nrf2–Gpx4 signaling pathway.	[81]
APS	DSS-induced colitis mice; Caco-2 cells	Inhibition	Astragalus polysaccharide prevents ferroptosis in a murine model of experimental colitis and human Caco-2 cells by inhibiting the NRf2–HO-1 pathway.	[84]
Curculigoside	DSS-induced colitis mice; IEC6-cells	Inhibition	Curculigoside inhibits ferroptosis in ulcerative colitis through the induction of GPX4.	[85]
RSL3	HCT116/LoVo/HT29 CRC cells	Induction	RSL3 drives ferroptosis through GPX4 inactivation and ROS production in colorectal cancer.	[6]
SLC7A11	HT29-cells	Induction	Targeting SLC7A11 specifically suppresses the progression of colorectal cancer stem cells by inducing ferroptosis.	[13]
Resibufogenin	HT29/SW480/NCM460 cells	Induction	Resibufogenin inhibits colorectal cancer cell growth and tumorigenesis by triggering ferroptosis and ROS production mediated by GPX4 inactivation.	[86]
MiR-19a	HT29-cells	Inhibition	MiR-19a suppresses the ferroptosis of colorectal cancer cells by targeting IREB2.	[90]
TIGAR	SW620/HCT116 cells	Induction	TIGAR drives colorectal cancer ferroptosis resistance through the ROS–AMPK–SCD1 pathway.	[92]
β-Elemonic acid	SW480/HCT116/HT29 cells; BALB/c nude mice	Induction	EA at high concentrations induces ferroptosis by downregulating FTL and upregulating TF, CP, and ACSL4.	[94]
ACSL4	I/R mice; Caco-2 cells	Induction	Ischemia-induced ACSL4 activation contributes to ferroptosis-mediated intestinal injury in intestinal ischemia–reperfusion.	[14]
Ionizing radiation	Balb/c mice	Induction	Ferroptosis can promote ionizing radiation-induced intestinal damage.	[100]
*P. aeruginosa*	Female C57BL/6 mice; Caco-2 cells	Induction	The colonization of the gut with *P. aeruginosa* induces ferroptosis in host cells and significantly reduces survival in irradiated mice.	[101]

### 3.2. Role of Necroptosis in Intestinal Diseases

#### 3.2.1. Necroptosis and IBD

A growing number of studies have shown that necroptosis plays a role in the development of intestinal inflammation. Patients with CD and ulcerative colitis have elevated levels of RIPK1, RIPK3, and MLKL in diseased tissues, suggesting that necroptosis is linked to an increased risk of IBD (Table 2). Supposedly, polysaccharides from edible mushrooms inhibit colitis, and polysaccharide extracts significantly inhibit the receptor-interacting protein kinase RIPK1–RIPK3–MLKL necroptotic signaling cascade, thereby decreasing the levels of phosphorylated MLKL in the mouse colon [106]. This phenomenon supports using polysaccharides as an alternative source of therapeutic agents for UC, wherein RIP3 overexpression promotes necroptosis-induced inflammation and disrupts intestinal epithelial barrier integrity [107]. On the other hand, the inhibition of the RIPK3 pathway reduces intestinal inflammation and cell death in IBD and inhibits the growth of peripheral mononuclear cells in patients with UC [107]. Therefore, the targeted inhibition of RIPK3 may be a novel approach for IBD treatment and has emerged as a promising candidate for IBD treatment in recent years. The traditional herbal formula Wu-Mei-Wan inhibits necroptosis in mice by increasing RIPK3 O-GlcNAcylation, thus alleviating TNBS-induced colitis [108]. In addition, the A20-binding inhibitor of NF-κB3 (ABIN3) negatively regulates necroptosis-induced intestinal inflammation by recruiting A20 and limiting the ubiquitination of RIPK3 in IBD [109]. Recent studies have shown that TNF-a induces LGR5D stem cell dysfunction in patients with Crohn’s disease, and exogenous prostaglandin E2 (PGE2) treatment restores LGR5+ stem cell function [110]. Therefore, PGE2 may be a promising therapeutic candidate for improving intestinal epithelial repair and regeneration in patients with CD. In addition, PGE2 signaling via type E prostaglandin receptor 4 (EP4) on intestinal epithelial cells (IECs) inhibits epithelial necroptosis and induces the regression of colitis [111]. Mechanistically, EP4 signaling on IEC clusters on RIPK1 inhibits TNF-induced necroptotic effects, mixed-spectrum kinase structural domain-like pseudokinase activation, and membrane translocation. These studies indicate that PGE2 could be a new target in the treatment of IBD.

#### 3.2.2. Necroptosis and CRC

Recent studies have suggested that tumor cells resistant to apoptosis may be sensitive to the necroptosis pathway, indicating that necroptosis and its regulatory mechanisms are potential targets for CRC therapy (Table 2). Han et al. [112] showed that resibufogenin induced CRC cell necrosis through RIPK3-mediated necroptosis, thus inhibiting tumor growth. In addition, fragile X mental retardation protein (FMRP) binds to *RIPK1* mRNA, and treatment with FMR1 anti-transcription upregulates RIPK1, leading to the necroptosis of CRC cells and the inhibition of tumor growth [113]. Some studies have shown that SET and MYND domain-containing protein 2 (SMYD2) targets RIPK1 and limits TNF-induced apoptosis and necroptosis to support colon tumor growth [114]. The A20-binding inhibitor of NF-κB1 (ABIN-1), also known as TNIP1, is a ubiquitin-binding protein that inhibits RIPK1-independent apoptosis, necroptosis, and NF-κB activation, and ABIN-1 deficiency enhances necroptosis-based CRC treatment [115]. Kushen injection is a commonly used adjuvant to CRC chemotherapy, and combined with cisplatin, it exerts synergistic antitumor activity against p53-R273H/P309S mutant CRC cells through the induction of apoptosis [116]. Chemoresistance to 5-Fu is common in CRC. Zhang et al. found that the induction of necroptosis by GDC-0326 is associated with regulating RIPK1 and RIPK3 and that GDC-0326 inhibits the growth of CRC cells in a dose-dependent manner [117]. Furthermore, GDC-0326 plus 5-Fu has enhanced antitumor efficacy and an acceptable safety profile and might be a promising therapeutic strategy for future CRC patients. This finding suggests that the promotion of necroptosis enhances the sensitivity of CRC to chemotherapeutic agents, thus contributing to the drug treatment of CRC. Furthermore, MLKL is a crucial mediator of necrosis, leading to the release of cell DAMPs, and *mlkl^−/−^* mice are susceptible to colitis and colitis-associated tumorigenesis [118]. Notably, the low expression of MLKL in human colorectal tumors enhances the activation of the signal transducer and activator of transcription 3 (STAT3). It is correlated with decreased overall survival, suggesting that MLKL inhibits colorectal carcinogenesis by suppressing the STAT3 signaling pathway [119].

#### 3.2.3. Necroptosis and Intestinal Injury

Necroptosis is related to the pathological conditions of intestinal I/R injury and is a potential target for treating the condition [120] (Table 2). DAMPs are released during intestinal I/R, leading to severe inflammatory responses. Mitochondrial DNA (mtDNA) is a DAMP that can be released during intestinal I/R. A previous study reported that mtDNA derived from intestinal epithelial cells exacerbates the inflammatory response and intestinal barrier dysfunction during intestinal I/R injury, but the exact mechanism is unclear [121]. Interestingly, the stimulator of interferon gene (STING) signaling is involved in mtDNA-induced necroptosis in intestinal I/R injury, while STING knockout ameliorates mtDNA-induced intestinal I/R injury necroptosis [122]. Li et al. [123] found that targeting the necroptosis of intestinal epithelial cells by Nec-1 alleviates intestinal injury after intestinal I/R in rats. Therefore, an in-depth study of necroptosis would be useful in planning the treatment of intestinal I/R. Sepsis can disrupt the intestinal mucosal barrier by releasing many inflammatory mediators. Liu et al. [124] injected *Escherichia coli* LPS into piglets to establish a sepsis model of intestinal injury during this period. Moreover, LPS activates the necroptosis signaling pathway in the gut, impairing gut morphology and function. Furthermore, the inhibition of necroptosis with Nec-1 attenuated LPS-induced intestinal damage, thereby suggesting that necroptosis contributes to LPS-induced intestinal injury and that Nec-1 exerts a preventive effect on an intestinal injury during sepsis. Previous studies have shown that, in a piglet model, flaxseed oil attenuates intestinal injury by downregulating LPS-induced necroptosis and TLR4/NOD signaling [125]. The intestinal mucosal barrier resists pathogen invasion and maintains intestinal homeostasis [126]. The necroptosis of IECs leads to bacterial translocation, and inflammatory cells release inflammatory factors, disrupting the integrity of the intestinal mucosal barrier. Dong et al. [127] demonstrated that SpvB induces necroptosis by downregulating RIPK3 degradation, ultimately destroying the intestinal epithelial barrier and aggravating intestinal injury. Heavy metals have toxic effects on intestinal epithelial cells. Some studies have shown that cadmium induces necroptosis in pigs by increasing ROS accumulation and Th1/Th2 imbalance, thereby aggravating minor intestinal injuries [128]. Nrf2 is a critical regulator of antioxidants and plays a key role in regulating intestinal damage. Xu et al. [129] found that Nrf2 attenuates radiation-induced rectal injury by negatively regulating the pRIP1–pRIP3–pMLKL-dependent necroptosis pathway, indicating that Nrf2 is a potential target for radiation-induced rectal injury therapy.

**Table 2 biomolecules-13-00820-t002:** Mechanisms of necroptosis in inflammatory bowel diseases, colorectal cancer, and intestinal injury.

Compound/Target	Model	Effect	Mechanism	Ref.
Wu-Mei-Wan	TNBS-induced colitis mice	Inhibition	Wu-Mei-Wan alleviates TNBS-induced colitis in mice by inhibiting necroptosis through increasing RIPK3 O-GlcNAcylation.	[108]
ABIN3	DSS-induced colitis mice; AOM/DSS mice	Inhibition	ABIN3 regulates the intestinal inflammatory response by interacting with A20 and regulating the K63 deubiquitination modification of necroptosis in IBD.	[109]
Resibufogenin	SW480/MC38 cells;The splenic capsule of mice	Induction	Resibufogenin inhabits CRC growth through RIP3-mediated necroptosis.	[112]
FMRP	WT and Fmr1 KO mice were injected with the AOM; CRC samples from patients	Induction	FMRP regulates RIPK1 and CRC resistance to necroptosis.	[113]
LPS	Weaned pigs	Induction	LPS activates the necroptosis signaling pathway in the gut and leads to impaired gut morphology and function.	[124]
Flaxseed Oil	Weaned pigs	Inhibition	Flaxseed oil attenuates intestinal damage and inflammation by regulating necroptosis and TLR4/NOD signaling pathways.	[125]
SpvB	C57BL/6J female mice; Caco-2 cells	Inhibition	SpvB accumulates a large amount of RIPK3 by inhibiting the degradation of RIPK3, which eventually leads to the aggravation of intestinal damage.	[127]
Cadmium	IPEC-J2 cells; weaned swine	Induction	Cadmium induces necroptosis in pigs by increasing ROS accumulation and Th1/Th2 imbalance, thereby aggravating small intestinal injuries in pigs.	[128]
Nrf2	C57BL/6 Nrf2 KO mice; human intestinal epithelial crypt (HIEC) cells	Inhibition	Nrf2 attenuates radiation-induced rectal injury by negatively regulating necroptosis.	[129]

### 3.3. Role of Pyroptosis in Intestinal Diseases

#### 3.3.1. Pyroptosis and IBD

Pyroptosis is a proinflammatory programmed cell death linked to various inflammatory conditions. It is mainly related to inflammatory bowel illness in the case of intestinal disorders (Table 3). The intestinal mucosa expresses caspase-11, preventing DSS-induced colitis [130]. One of the most critical components of the inflammatory response is the inflammasome, and the dysregulation of NLRP3 inflammasomes is associated with IBD [59]. Deng et al. [131] found that NR4A1 inhibits the NLRP3 inflammasome to prevent inflammation and protect against *C. rodentium*-induced colitis. Nek7 is an essential component of NLRP3 inflammasome activation and coordinates with NLRP3 to regulate pyroptosis in IBD by activating NF-κB signaling [132]. Recently, JQ1 pretreatment was found to reverse high levels of proinflammatory cytokines, IL-6, IL-1b, and IL-18, in the endotoxemic colon and to reduce the levels of phosphorylated NF-κB and NLRP3/ASC/caspase 1 inflammasome complexes [133], suggesting that JQ1 is a promising target for the treatment of colitis. Exosomes have been implicated in various diseases. A recent study showed that hucMSC-derived exosome miR-378a-5p targets NLRP3, leading to the blockage of NLRP3 inflammasome assembly and subsequent caspase-1 cleavage [134]. This reduces GSDMD pore formation and improves IBD. These studies suggested that NLRP3 may be a promising therapeutic target for IBD. Pyroptosis has been identified as gasdermin-mediated proinflammatory cell death. GSDME^−/−^ mice exhibited milder intestinal inflammation than wild-type (WT) mice [135], suggesting that the inhibition of GSDME-mediated proptosis with drug targeting may provide a functional therapeutic approach for CD treatment. The above studies indicate that the inhibition of pyroptosis reduces IBD-induced damage. Therefore, the targeted inhibition of excessive inflammatory activation caused by pyroptosis could be a promising therapeutic approach for IBD.

#### 3.3.2. Pyroptosis and CRC

GSDMD-dependent pyroptosis, including CRC, is closely associated with tumor development (Table 3). Secoisolariciresinol diglucoside (SDG) inhibits the growth of CRC cell lines related to the induction of GSDMD-dependent pyroptosis by SDG by generating the ROS–P13K–AKT–BAK–mitochondrial apoptotic pathway [136]. Nanostructured toxins can be used to selectively destroy drug-resistant human CXCR4(+) CRC stem cells [137]. Furthermore, LPS enhances oxaliplatin chemosensitivity to HT29 cells via GSDMD-mediated pyroptosis [138], which provides a reliable therapeutic target against resistance to CRC chemotherapeutic agents. However, many studies have found that GSDMD is not the only enforcer of pyroptosis. GSDME-induced pyroptosis also plays a critical role in CRC [139]. In the colitis-associated colorectal cancer (CAC) model, Gsdme^−/−^ mice showed reduced weight loss and colonic shortening compared with WT mice. Treatment with neutralizing anti-high-mobility group protein 1 (HMGB1) antibody reduced the number and size of tumors, ERK1/2 activation, and proliferating cell nuclear antigen expression in azoxymethane-/DSS-treated WT mice, suggesting that GSDME-mediated pyroptosis promotes CAC development by releasing HMGB1 [140]. In addition, GSDME knockdown shifted apoptosis-induced cell death from pyroptosis to apoptosis in vitro, suggesting that apoptosis induces CRC pyroptosis via a GSDME-dependent pathway [141]. As mentioned above, GSDMDE plays a central role in the regulation of pyroptosis. In addition to the regulators mentioned above, novel compounds, such as GW4064, potentially promoting pyroptosis and inhibiting CRC have been identified [142]. Researchers have also identified novel targets regulating CRC pyroptosis, including A438079 [143]. In conclusion, these investigations elucidate the putative processes and role of pyroptosis in CRC. Thus, these findings might provide an excellent strategy to improve the treatment and prognosis of this cancer.

#### 3.3.3. Pyroptosis and Intestinal Injury

Pyroptosis releases a large number of inflammatory cytokines and DAMPs, leading to intestinal epithelial defects. Currently, studies have shown the existence of pyroptosis in intestinal I/R injury (Table 3). It has been reported that caspase-1-related pyroptosis promotes acute lung injury after intestinal I/R injury [144]. Jia et al. [145] found that metformin treatment protects the intestinal barrier function from intestinal I/R injury and reduces oxidative stress and inflammatory responses by modulating pyroptosis. Sepsis-induced intestinal injury is associated with pyroptosis. Zhang et al. [146] demonstrated that cecal ligation and puncture-induced pyroptosis are elevated but inhibited via CB2 agonist HU308 treatment due to a decreased level of NLRP3 protein and activated caspase-1 and GSDMD. Together, these phenomena indicate that CB2 receptor activation is mediated by the inhibition of pyroptosis, which exerts a protective role in sepsis-induced intestinal injury. Dihydromyricetin attenuates ileal mucosal damage, oxidative stress, and pyroptosis; maintains intestinal barrier function; and inhibits the LPS-triggered NLRP3 inflammasome and the activation of the TLR4–NF-kB signaling pathway [147]. Several compounds can modulate pyroptosis in intestinal injury. Berberine inhibits Wnt/b-catenin pathway activation by modulating the miR–103a–3p–BRD4 axis, inhibits pyroptosis, and reduces the intestinal mucosal barrier defects caused by colitis [148]. Schisandrin B reduces ROS-induced mitochondrial damage by inhibiting pyroptosis and epithelial cell damage in colitis [149]. Although radiation therapy is a crucial clinical treatment option for cancer, the small intestine is highly sensitive to radiation and can be damaged by radiation [150]. Therefore, finding protective strategies against radiation-induced intestinal damage is essential. Another study found that CRC barely expressed GSDME, while GSDME was significantly expressed in the surrounding normal intestinal tissue, suggesting that radiation-induced intestinal damage is regulated by GSDME [151]. Recent studies have indicated that polydopamine nanoparticles inhibit pyroptosis and attenuate ionizing radiation-induced intestinal damage, deeming them potential radioprotectants [152]. A recent study reported that p-coumaric acid ameliorated ionizing radiation-induced intestinal damage by modulating oxidative stress, inflammation, and pyroptosis [153]. Therefore, these findings might provide a potential treatment strategy to attenuate the intestinal injuries induced by intestinal I/R, sepsis, and radiation therapy.

**Table 3 biomolecules-13-00820-t003:** Mechanisms of pyroptosis in inflammatory bowel diseases, colorectal cancer, and intestinal injury.

Compound/Target	Model	Effect	Mechanism	Ref.
NR4A1	Marrow-derived macrophages (BMDMs)from WT mice and NR4A1^−/−^ mice	Inhibition	NR4A1 inhabits pyroptosis by transcriptionally inhibiting NLRP3 and IL-1β and colocalizing with NLRP3 in trans-Golgi to alleviate pathogenic bacteria-induced colitis.	[131]
NEK7	MODE-K cells; DSS-induced colitis mice	Induction	NEK7 interacts with NLRP3 to regulate pyroptosis in IBD via NF-κB signaling	[132]
JQ1	LPS-induced endotoxemia mice	Inhibition	JQ1 blocks inflammatory pyroptosis-related acute colon injury induced by LPS.	[133]
hucMSC-derived exosomes	DSS-induced colitis mice	Inhibition	hucMSC-derived exosomes attenuate colitis by modulating macrophage pyroptosis via the miR–378a–5p–NLRP3 axis.	[134]
SDG	HCT116 cells	Induction	SDG induces pyroptosis by activating caspase-1 to cleave GSDMD in CRC cells.	[136]
Nanostructured toxins	CRC samples from patients	Induction	Nanostructured toxins for the selective destruction of drug-resistant human CXCR4(+) colorectal cancer stem cells.	[137]
GW4064	HT-29, SW480/HCT1 16/CACO-2/RKO cells	Induction	GW4064 enhances the chemosensitivity of colorectal cancer to oxaliplatin by inducing pyroptosis.	[142]
A438079	HCT-116/SW620 cells; xenografted mice	Induction	A438079 affects CRC cell proliferation, migration, apoptosis, and pyroptosis by inhibiting the P2X7 receptor.	[143]
Metformin	I/R mice; Caco-2 cells	Inhibition	Metformin treatment protects gut barrier function from intestinal I/R injury and reduces oxidative stress and inflammatory responses by modulating pyroptosis.	[145]
CB2 receptor	LPS induced-sepsis mice	Inhibition	CB2 receptor activation plays a protective role in sepsis-induced intestinal injury by inhibiting pyroptosis.	[146]
Berberine	DSS-induced colitis mouse; Caco-2 cells	Inhibition	Berberine negatively regulates the Wnt/β-catenin pathway, thereby attenuating colitis-stimulated pyroptosis and intestinal mucosal barrier defects.	[148]
GSDME	CT26 cells; GSDME-KO CT26 cells	Induction	GSDME enhances radiation-induced pyroptosis in CRC cells and normal epithelial cells via a caspase-3-dependent pathway.	[151]
Dopamine-derived nanoparticles	X-ray-induced mice	Inhibition	PDA-NPs inhibit pyroptosis and alleviate ionizing radiation-induced intestinal damage.	[152]
p-Coumaric acid	X-ray-induced mice	Inhibition	p-Coumaric acid (CA) ameliorates ionizing radiation-induced intestinal damage by modulating oxidative stress, inflammation, and pyroptosis.	[153]

## 4. Conclusions and Perspectives

Ferroptosis, necroptosis, and pyroptosis are newly identified types of programmed cell death associated with the development of intestinal diseases. The inhibition of ferroptosis, necroptosis, and pyroptosis attenuates intestinal I/R, sepsis, as well as radiation-induced IBD and intestinal injury. In contrast, pharmacological activators that induce ferroptosis, necrosis, and pyroptosis inhibit CRC migration, invasion, and proliferation, suggesting their dual roles in various intestinal diseases. Additional compounds and targets associated with ferroptosis, necroptosis, and pyroptosis in intestinal disease have been identified. However, whether specific regulators or signaling pathways are involved in these diseases remains unclear. Interestingly, many intestinal diseases are associated with one or more cell death processes, including other modes not mentioned in this review, such as apoptosis, autophagy, and necrosis. Therefore, further studies are needed to identify intestinal disease-specific cell death mechanisms for developing disease context-relevant therapeutic options. Future studies should focus on the crosstalk between cell death in intestinal diseases.

Although many advances have been made in understanding the mechanisms and roles of ferroptosis, necroptosis, and pyroptosis in intestinal diseases, further clinical applications of targeted therapies for ferroptosis, necroptosis, and pyroptosis are still at an early stage. Their specific role remains to be investigated across the spectrum of intestinal diseases, including many not covered in this review. Therefore, we believe that more studies, including animal experiments and clinically relevant studies, are needed to understand the role of different processes of cell death in intestinal diseases in more detail so that new clinical treatments can be provided for intestinal diseases.

## Figures and Tables

**Figure 1 biomolecules-13-00820-f001:**
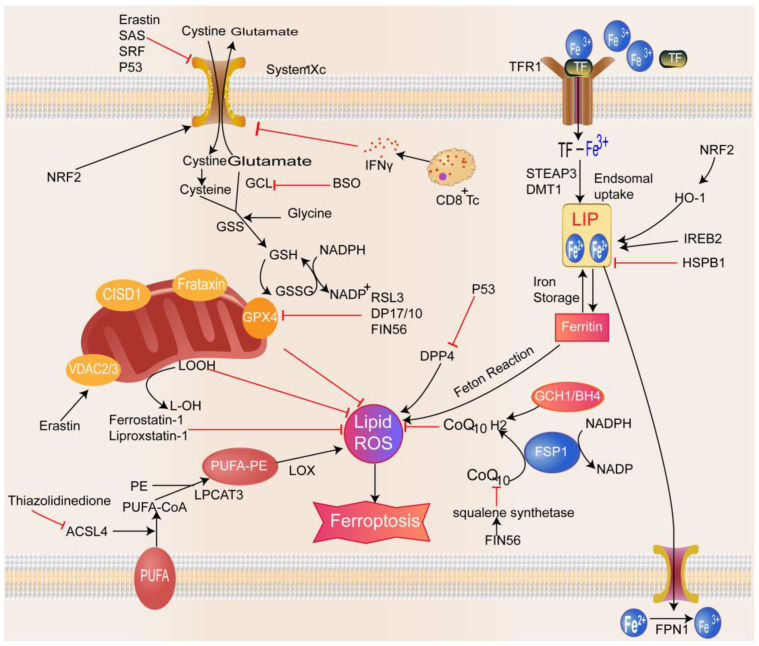
Mechanisms of ferroptosis. Ferroptosis is mainly related to disorders of amino acid metabolism, accumulation of lipid peroxides, and disorders of iron ion metabolism. Xc complex imports cystine for the synthesis of glutathione. GPX4 uses glutathione to prevent the accumulation of lipid-reactive oxygen species. The classical Fenton reaction between Fe^3+^ and Fe^2+^ produces abundant reactive oxygen species. Decreased iron stores or increased iron intake can lead to iron overload and eventually iron death. In addition, other signaling pathways and regulators control ferroptosis sensitivity. For example, erastin can bind to porin 2/3 on the outer mitochondrial membrane, causing mitochondrial dysfunction and the release of many oxidative substances, ultimately leading to ferroptosis.

**Figure 2 biomolecules-13-00820-f002:**
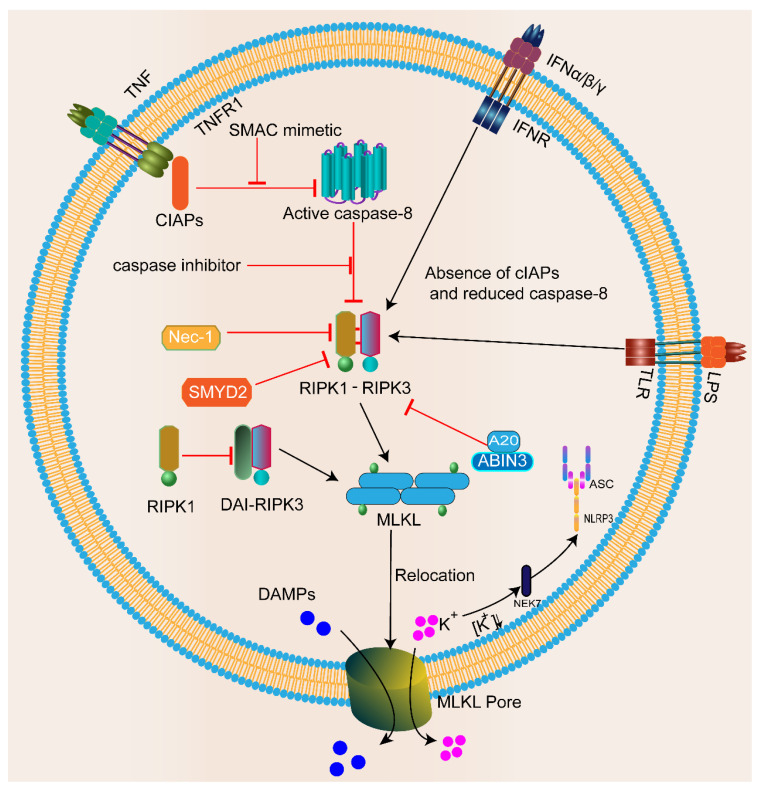
Mechanisms of necroptosis. Death receptors (TNFR, TLR, and IFNR) bind to their corresponding ligands (as shown) and are activated to trigger necroptosis. Upon caspase-8 or cIAP depletion, they promote the assembly of the RIPK1–RIPK3–MLKL signaling complex, resulting in the phosphorylation of MLKL (p-MLKL). Phosphorylated MLKL translocates to the plasma membrane to initiate membrane damage and form macropores. Ultimately, MLKL pores lead to necroptosis by allowing ion influx, cell swelling, membrane lysis, and subsequent uncontrolled release of intracellular substances. Due to membrane damage, potassium efflux can further activate NLRP3 through NEK7, increasing the release of inflammatory mediators. Recent studies have also found inhibitory factors of necroptosis, such as Nec-1 and SMYD2.

**Figure 3 biomolecules-13-00820-f003:**
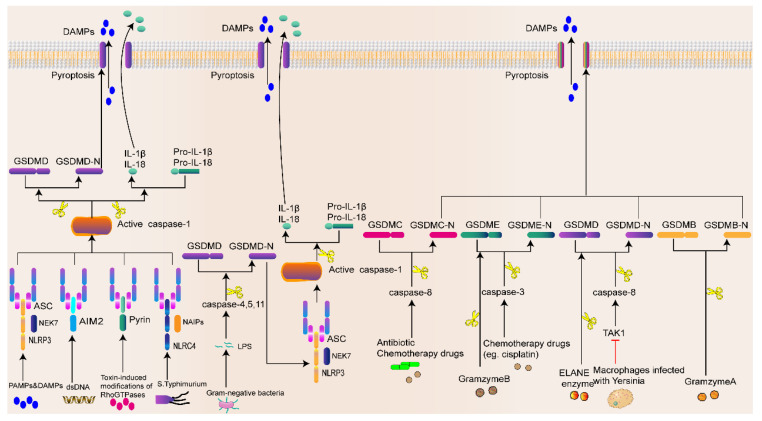
Mechanisms of pyroptosis. Sensing pathogens or DAMPs by inflammatory vesicles promotes the formation of an inflammatory vesicle complex that involves NLRP3/NLRC4/AIM2/Pyrin, ASC, and caspase-1 in the classical pathway. In the nonclassical pathway, LPS leads to caspase-11 activation. Activated caspase-1 and caspase-11 shear GSDMD and form GSDMD-NT. Some chemotherapeutic agents can induce the activation of caspase-3 and caspase-8. Then, activated caspase-3 and caspase-8 can cleave GSDME and GSDMC and form GSDME-NT with GSDMC-. In mouse macrophages, Yersinia pestis infection inhibited TAK1 activity, which caused the cleavage of GSDMD by caspase-8. GSDMB-NT, GSDMC-NT, GSDMD-NT, and GSDME-NT can oligomerize and translocate to the plasma membrane, thus forming cellular pores, causing cytoplasmic swelling, and releasing proinflammatory cytokines and cellular contents.

## Data Availability

Not applicable.

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
