# Peer review of "The Induction Mechanism of Ferroptosis, Necroptosis, and Pyroptosis in Inflammatory Bowel Disease, Colorectal Cancer, and Intestinal Injury"

_biomolecules, 2023, doi:10.3390/biom13050820_

Round 1

Reviewer 1 Report

This is  an  interesting  manuscript attempting  to  address  mechanisms  of  cell death  and  intestinal diseases. 

1)     Specifying  which diseases  in title  will facilitate  searches

2)     There are  numerous recent reviews  on  cell death mechanisms,  I would  give a  summary of  each one and  refer  to  current  literature, which  is extensive

3)     Line  57  what are  intracellular  substances

4)     Line  63 “life- and health-threatening” simplify to  health- threatening

5)     Line  6-67  This  paragraph  lacks  focus.. too many  ideas at once.

6)     Line  91  There are  many other Xc- regulators

7)     Role  of  mitochondria..?

8)     Possible  mistake  in fig 1 right below  Xc diagram”cysteine”

9)     The authors organize tables  by health, perhaps  suborganize (shaded area  for example)  by  mechanism

Author Response

Point 1: Specifying which diseases in title will facilitate searches.

Response: Thank you for your very good suggestion. We strongly agree with the suggestion to specify the disease in the title so that it can be useful for retrieval. We have therefore revised the title to Induction mechanism of ferroptosis, necroptosis, and pyroptosis in inflammatory bowel disease, colorectal cancer, and intestinal injury.

Point 2: There are numerous recent reviews on cell death mechanisms, I would give a summary of each one and refer to current literature, which is extensive.

Response: Thank you for your very good comment. At present, there are indeed many reviews on the mechanism of cell death, which mainly discuss the mechanism of cell death in inflammatory diseases and tumors[1-4]. In terms of intestinal diseases, most recent reviews have focused on the mechanisms of cell death in inflammatory bowel disease and colorectal neoplasms[5-7]. Among them, ferroptosis, necroptosis, and pyroptosis can induce or aggravate inflammatory bowel disease but play an antitumor role in colorectal cancer. However, many recent studies have found that cell death plays a key role in intestinal injury. Therefore, in our review, we also focused on the role of ferroptosis, necroptosis, and pyroptosis in intestinal injury caused by intestinal ischemia reperfusion, sepsis, and radiation therapy. We concluded that ferroptosis, necroptosis, and pyroptosis can aggravate intestinal injury caused by intestinal ischemia reperfusion, sepsis, and radiation therapy. In addition, we summarize the molecular targets and compounds related to ferroptosis, necroptosis, and pyroptosis in intestinal diseases to provide new ideas for the clinical treatment of related intestinal diseases. (Please see Page 8~16, table 1~3).

Point 3: Line 57 what are intracellular substances.

Response: Thank you for your reminder. We must apologize for that the initial manuscript did not clearly clarify the expression. We have modified the description. (Please see Page 2, margins 55~58).

Point 4: Line 63 “life-and health-threatening” simplify to health- threatening

Response: Thank you for your suggestion. Since this paragraph lacked focus, we have modified the paragraph and removed this content. (Please see Page 2, margins 61~73).

Point 5: Line 6-67 This paragraph lacks focus.. too many ideas at once.

Response: Thank you for your good suggestion. We have modified this paragraph to focus on intestinal epithelial cell death, suggesting a link between ferroptosis, necroptosis, and pyroptosis and the development of intestinal disease. (Please see Page 2, margins 61~73).

Point 6: Line 91 There are many other Xc- regulators.

Response: Thank you for your reminder. Xc- is an essential target in the regulation of ferroptosis. Studies have shown that erastin, sulfasalazine (SAS), and sorafenib can bind to related transport proteins, block the transport function of system Xc- and induce cellular ferroptosis[8-10]. (Please see Page 3, margins 97~101).

Point 7: Role of mitochondria...?

Response: Thank you for your good comment. Many studies in recent years have shown that mitochondria play a role in both promoting and inhibiting ferroptosis. On the one hand, the promoting role is related to voltage-dependent anion channels and cardiolipin. On the other hand, the inhibitory role is associated with CDGSH iron-sulfur domain 1 and mitochondrial ferritin[11-14]. (Please see Page 4, margins 160~171).

Point 8:  Possible mistake in fig 1 right below Xc diagram “cysteine”

Response: Thank you for your reminder. We have found a spelling error in “cysteine” and corrected it. (Please see: Page 3, figure 1).

Point 9: The authors organize tables by health, perhaps suborganize (shaded area for example) by mechanism.

Response: Thank you for your good suggestion.We totally agree with the reviewer's opinion. We have recreated the table according to the mechanism. (Please see: Page 8~16, table 1~3).

Reviewer 2 Report

The manuscript #biomolecules-2237397 has been carefully reviewed. Since the role and mechanism of ferroptosis, necroptosis, and pyroptosis have been discussed in several published manuscripts, it would be great to describe in detail the advantages of this review compared to others. What gaps does this review article cover?

In addition, the "conclusions and perspectives" section needs to be rewritten with more ideas and discussion instead of presenting the questions. Posing the challenging questions is great but at the end of a review article, the audience wants to read some specific leading points and the general restating of information is not satisfactory in this section.

Author Response

Point 1: The manuscript #biomolecules-2237397 has been carefully reviewed. Since the role and mechanism of ferroptosis, necroptosis, and pyroptosis have been discussed in several published manuscripts, it would be great to describe in detail the advantages of this review compared to others. What gaps does this review article cover?

Response: Thank you for your good suggestions. Recently, the physiological and pathological effects of cell death in various diseases have been extensively studied[15-17]. However, there are still gaps in the basic knowledge of the mechanism of intestinal epithelial cell death in intestinal diseases. It is important to note that ferroptosis, necroptosis, and pyroptosis are indispensable during intestinal development and homeostasis and occur throughout life. Therefore, we review the ferroptosis, necroptosis, and pyroptosis implicated in the regulation of intestinal diseases and highlight their underlying molecular mechanisms in light of potential therapeutic applications to develop emerging and promising therapeutic strategies. We have added these comments to the introduction section. (Please see Page 2, margins 61~73).

Point 2: In addition, the "conclusions and perspectives" section needs to be rewritten with more ideas and discussion instead of presenting the questions. Posing the challenging questions is great but at the end of a review article, the audience wants to read some specific leading points and the general restating of information is not satisfactory in this section.

Response: Thank you very much for your suggestion. Based on your very helpful suggestions, we have rewritten the "Conclusions and perspectives" section. We present some specific points and additional thoughts and discussions to draw attention and thinking about the role of ferroptosis, necroptosis, and pyroptosis in intestinal diseases. (Please see Page 17, margins 533~554).

Reviewer 3 Report

The review by P. Zhou et al. represents a comprehensive analysis of several non-apoptotic modes of cell death in the intestine. The authors dissected general definitions of ferroptosis, pyroptosis and necroptosis and then reported a vast body of literature on these mechanisms as factors of disease as well as tissue response to treatment. The review is a detailed study expected to serve as a scholarly compendium. 

Two issues should be addressed before the study can be recommended for publication:

1. What is the relation of the analyzed non-apoptotic mechanisms to apoptosis? Is there indeed a strict demarcation line? Sometimes caspase activation and mitochondrial events, typical phenomena in apoptosis, occur in 'non-apoptotic' situations. Definitely, the involvement of individual mechanisms is tissue- and stimulus specific. For example, ferroptosis is special and perhaps indispensable in particular sitiations; however, cell death induced by anticancer drugs or radiation is a balance of several phenomena.  What is the authors' opinion in regard to the complexicity of death mechanisms and the roles of individual death modes in clinical settings?

2. A thorough linguistic editing is worthy, I mean grammar and scientific essence. I recommend to avoid trivial statements and an unnecessary generalization. It is at the discretion of the authors to select only the most significant literature sources with more focus on clinical analysis.

Author Response

Point 1: What is the relation of the analyzed non-apoptotic mechanisms to apoptosis? Is there indeed a strict demarcation line? Sometimes caspase activation and mitochondrial events, typical phenomena in apoptosis, occur in 'non-apoptotic' situations. Definitely, the involvement of individual mechanisms is tissue- and stimulus specific. For example, ferroptosis is special and perhaps indispensable in particular sitiations; however, cell death induced by anticancer drugs or radiation is a balance of several phenomena. What is the authors' opinion in regard to the complexicity of death mechanisms and the roles of individual death modes in clinical settings?

Response: Thank you very much for your good comments. Ferroptosis is a unique form of iron-dependent nonapoptotic regulated cell death that is pathologically characterized by the accumulation of intracellular iron-dependent lipid peroxidation[10]. Necroptosis is characterized by a serine/threonine protein kinase 1/3 (RIPK1/RIPK3)-mediated phosphorylation signaling pathway activating mixed lineage kinase domain-like protein (MLKL/pMLKL) [18]. Pyroptosis is a form of programmed cell death that accompanies inflammation [19]. It is characterized by gasdermin family proteins (GSDMs) mediating cellular pore formation, causing cell swelling and lysis, releasing cellular contents such as proinflammatory cytokines and DAMPs, and activating a robust inflammatory response[20]. Apoptosis is a strictly regulated form of programmed cell death carried out by multicellular organisms as part of normal development. The two main pathways of apoptosis are intrinsic and extrinsic[21]. Mitochondria are double-membrane bound organelles that not only provide energy for intracellular metabolism but also play a key role in a variety of processes regulating cell death[22]. As the main site of iron utilization and the main regulator of oxidative metabolism, mitochondria are the main source of reactive oxygen species (ROS), and many studies in recent years have shown that mitochondria play a role in promoting and inhibiting ferroptosis[11-14]. However, studies have shown that mitochondrial ROS promote susceptibility to infection through gasdermind-mediated necroptosis[23]. In addition, mitochondrial ROS have been shown to promote pyroptosis of macrophages by inducing GSDMD oxidation[24]. Pyroptosis and apoptosis are two forms of regulated cell death driven by active caspases accompanied by permeabilization of the outer mitochondrial membrane. Gasdermins mediate the cellular release of mitochondrial DNA during pyroptosis and apoptosis[25]. Thus, mitochondrial oxidation is a key event in the crosstalk among ferroptosis, necroptosis, and pyroptosis. It has been shown that BAX and BAK activate caspase-3 and -7 and subsequently induce NLRP3 inflammasome activation and IL-1β secretion by inducing potassium efflux[10]. Notably, caspase-8, another apoptosis effector, was shown to have the dual functions of promoting and inhibiting NLRP3 inflammasome activation[26-28]. GPX4 is an important negative effector of ferroptosis and has recently been shown to inhibit caspase-11-dependent pyroptosis and IL-1β release[29]. It is increasingly clear that there is no strict boundary between apoptosis and nonapoptosis. Therefore, a better understanding of the physiological and mechanistic aspects of cell death signaling will provide a rationale for the clinical treatment of relevant diseases. We believe that future research should focus on the crosstalk between cell death to develop effective disease treatments. (Please see Page 17, margins 533~554).

Point 2: A thorough linguistic editing is worthy; I mean grammar and scientific essence. I recommend to avoid trivial statements and an unnecessary generalization. It is at the discretion of the authors to select only the most significant literature sources with more focus on clinical analysis.

Response: Thank you very much for your helpful suggestion. This manuscript has been edited by a native English-speaking colleague to remove unnecessary generalizations and statements. The important literature sources are refined, and the relevant clinical analysis is expanded.

Round 2

Reviewer 1 Report

The authors addressed most of  the  points in a satisfactory  manner

Reviewer 3 Report

The authors addressed the reviewers' comments. Now the study can be accepted for publication in Biomolecules. Nevertheless, a thorough lingusitic editing is needed.